# High-Throughput Chemometric Quality Assessment of Extra Virgin Olive Oils Using a Microtiter Plate Reader

**DOI:** 10.3390/s19194169

**Published:** 2019-09-26

**Authors:** Huihui He, Weiying Lu

**Affiliations:** Institute of Food and Nutraceutical Science, School of Agriculture and Biology, Shanghai Jiao Tong University, Shanghai 200240, China

**Keywords:** high throughput microtiter plate reader, UV–Vis spectra, chemometrics, extra virgin olive oil quality assessment

## Abstract

A commercially available microtiter plate reader was applied as a high-throughput counterpart of ultraviolet-visible (UV–Vis) spectrophotometer to identify the producing location of extra virgin olive oils (EVOOs). Multiplicative scatter correction and the first derivative was used to denoise the UV–Vis spectra and eliminate the effects of background drift. The spectra were analyzed using chemometrics methods including the principal component analysis (PCA) and the partial least squares-discriminant analysis (PLS-DA). The PLS-DA model on full spectra using 5 latent variables showed a classification accuracy of 97.92% by cross-validation. The overall results demonstrated that the use of a UV–Vis spectrophotometer based on the microtiter plate reader combined with chemometrics can be applied to the quality assessment of EVOOs. It is demonstrated that the microtiter plate reader can be a high-throughput tool in the quality assessment of food ingredients.

## 1. Introduction

Olive oil comprises about 4% of the total vegetable oil production [1]. It is universally regarded as a symbol of the Mediterranean diet. Studies have confirmed its protective effect in the neurovascular and cardiovascular systems [2], such as lowering the risk of Alzheimer’s Disease, Parkinson’s disease [3,4,5] and coronary artery disease [6]. The functional properties of olive oil are inseparable from its functional components, the nutrigenomics data on which are still sparse, but increasing rapidly [7].

Because of the increasing popularity of the Mediterranean diet and remarkable nutritional properties, olive oil consumption and production had been expanding to nontraditional producing countries during the past two decades, which may result in inferior quality [1]. This globalization of the olive oil cultivation and trade also imposed regulations on the quality management of the whole production process [8]. Three of the most important standards are those specified by the European Union (EU), the International Olive Council and the Codex Alimentarius [9]. For example, the EU has introduced specific regulations (EU Regulation 2081/1992 and 510/2006) to protect the geographical origin of olive oil [10,11]. Meanwhile, geographic association and traceability of food products are highly valued quality cues by consumers who possess certain socio-economic and demographic characteristics that form distinct and clearly-defined market segments [12].

The classical chemical analysis procedures, including the analyses of fatty acids, polyphenol and phenolic acid, have been reported as sources of analytical data to establish olive oil region discrimination using chemometrics patterns [10,13]. However, the classical chemical analysis requires destructive sample pretreatment. In comparison, the spectroscopic methods such as ultraviolet–visible (UV–Vis), near-infrared (NIR) and mid-infrared (MIR) spectroscopies are advantageous for their simplicity, rapidity, non-destructiveness, and low cost. As a result, they may be suitable for fingerprinting. These spectroscopic techniques have been individually and jointly applied in determination of olive oil geographic origins [8,10,14].

The fingerprinting methods are usually combined with chemometrics to resolve the complex and convoluted nature of spectroscopic fingerprint data. Chemometrics involves a wide range of techniques. Specifically, preprocessing, classification, and clustering have been widely applied in spectroscopic data modelling, and are able to determine which grade or what producing location the commodity belongs to for the quality assessment of food. For instance, Forina et al. combined artificial nose, NIR and UV–Vis spectroscopy, using stepwise-linear discriminant analysis (STEP-LDA) as a feature selection technique, and built a model with linear discriminant analysis (LDA) and quadratic discriminant analysis-UNEQual dispersed classes (QDA-UNEQ) to verify the protected designation of origin (PDO) of Italian olive oil [15]. Alves et al. combined UV–Vis measurements and independent component analysis (ICA) to identify that foodstuff was from an alleged origin [16]. Borras et al. used headspace-mass spectrometry (HS-MS), MIR and UV–Vis in combination with a partial least squares-discriminant analysis (PLS-DA) to classify virgin olive oil samples [17].

The nondestructive, fast and real-time UV–Vis spectroscopic method may constitute one of the most promising and effective detection methods for olive oil quality control. However, the shortcomings of its complicated operation are prominent when the number of samples expands greatly. In particular, it requires manual operation in loading the sample into cuvette. Secondly, the cuvette must be cleaned between each measurement. The residue may increase the deviations of measurements. As a result, traditional commercially available benchtop UV–Vis spectrophotometer models may not be able to meet the need without modification. Increasing sample throughput and incorporating automation are needed.

Unlike the traditional UV–Vis spectrophotometer, the multi-row porous characteristic of multi-well plate furnishes the microtiter plate reader with the ability to detect samples in a high-throughput, fully automatic manner. The microtiter plate reader is also a popular and versatile instrument for a conventional biochemistry laboratory, because it offers flexibility and ease of use over a broad range of applications, such as enzyme-linked immunosorbent assays, protein detections and drug screening assays. Measurements performed by the microtiter plate reader save time and effort of sample-loading, as well as decreasing possible deviations among parallel groups.

In this work, a high-throughput determination of the geographic origin of extra virgin olive oil (EVOO) by the microtiter plate reader instead of a benchtop UV–Vis instrument were tested. The dataset was processed by chemometrics modelling to demonstrate a rapid method for quality assessment of EVOO. The classical UV–Vis measurements using a benchtop model were also performed and compared. This work will help to extend the capabilities of the microtiter plate reader combined with chemometrics modelling in the quality and safety assessment of food ingredients.

## 2. Materials and Methods

### 2.1. EVOO Samples

Twelve commercially available EVOO samples produced from three countries including Spain, Italy and Turkey were purchased from local Chinese grocery stores. For each region, the sample set consisted of EVOO with four different brands. Specifically, Samples S1–S4, I1–I4, and T1–T4 were from Spain, Italy and Turkey, respectively.

Eight commercially available vegetable oils (two corn, four sunflower and two soy oils) were purchased from local Chinese grocery stores. All samples were tested directly without further treatments.

### 2.2. Experimental Section

#### 2.2.1. Benchtop UV–Vis Spectrophotometer

UV–Vis spectra data were collected at room temperature (25 °C) within 230–1000 nm, at 1 nm resolution by an Evolution 300 UV–Vis spectrophotometer (Thermo Fisher Scientific, Waltham, MA, USA). The radiation source was a xenon flash lamp. All samples were analyzed in triplicate in a rectangular quartz cuvette with a path length of 10 mm. The sample volume required for each measurement is around 3.5 mL. The UV–Vis data of the samples were compared to the microtiter plate reader assay after spectral baseline adjustment.

#### 2.2.2. Microtiter Plate Reader Assay

Samples were weighted at 200 mg in quintuplicate and placed into Nunc MicroWell transparent 96-well plates (Thermo Fisher Scientific). All samples were analyzed by an Infinite M1000 PRO microtiter plate reader (Tecan Group Ltd., Männedorf, Switzerland) at room temperature. Using a Quad4 monochromator, the microtiter plate reader gives flexibility of wavelength selection from UV to NIR. The radiation source was a xenon arc discharge lamp. The wavelength was set at 230–1000 nm with a 1 nm resolution, and the number of flashes was 25. The MTP200 dataset is then obtained from the spectra.

To determine whether the sampling amount is adequate, the microtiter plate reader assay procedure was also repeated at the weight of 100, 150 and 300 mg, and obtained datasets were labelled as MTP100, MTP150 and MTP300, respectively. The density of each sample was measured by using a DDM 2910 densimeter (Rudolph Research Analytical, Hackettstown, NJ, USA). Because the bands between 230 and 307 nm were saturated in the MTP assay, they were removed from all the spectra and only the data ranging in 308–1000 nm were analyzed. The datasets consist of a 60 × 693 matrix, where the rows represent the analyzed samples and the columns represent the variables.

### 2.3. Chemometrics Methods

The particles and turbidity of EVOO and variations in the optical path length can affect the UV–Vis spectra [8]. Therefore, the UV–Vis spectra detected by the microtiter plate reader need to be pre-processed before modelling to minimize the influence of noises caused by light scattering, baseline drift and other physical phenomena. The multiplicative scatter correction (MSC) is a normalization method to correct for scaling and offset variations in a set of spectra [18]. In the MSC method, every spectrum is considered to be linear with the ideal spectrum. The true ideal spectrum is not available and the average spectrum of sample set is often approximately estimated to be the ideal spectrum. The model used in the regression is as follow: $A_{i}=m_{i}{\bar{A}}_{i}+b_{i}$, in which ${\bar{A}}_{i}$ is the average spectrum of the sample set. The $m_{i}$ and $b_{i}$ coefficients are estimated by ordinary least squares regression over the different wavelengths. The light scattering corrected spectrum is calculated by
$$A_{i\left(MSC\right)}=\left(A_{i}-b_{i}\right)/m_{i}.$$

In addition to MSC, derivative methods are often used for processing one-dimensional signals to eliminate the effects of background drift. The first-level derivative eliminates the background constant and the second-level derivative eliminates the linear background shift. If the sampling wavelength intervals are equal, the first-level derivatives can be expressed as
$$y_{i}=x_{i+1}-x_{i}.$$

Here $y_{i}$ represents the series of derivatives of spectrum $x_{i}$. In this study, MSC followed by the first-level derivative (MSC-Der1) is applied.

Classification is involved in both ‘supervised’ and ‘unsupervised’ methods. Supervised methods extract general principles from observed examples guided by a specific prediction objective [19]. A set of data describing oils of known origin is used to construct models that are then used to classify geographically unknown oil samples into a previously constructed group [20]. Unsupervised methods group the data of oil samples that has not been classified or categorized.

This study focuses on two multivariate analysis methods: the unsupervised principal component analysis (PCA) and the supervised partial least squares-discriminant analysis (PLS-DA). The PCA is a widely used qualitative method in spectral data analysis [21]. The PCA reduces the dimensionality of numerical datasets, transforms the original variables into a series of new variables (principal components, PCs). PCs are linear combinations of the original variables. The first PC retains the maximum of the total variance and is followed by the second PC, and so forth. The projection of each sample in the new axis is called score, of which the biplots are usually used to present the sample distribution in a score space [22].

The PLS-DA combines the properties of partial least squares regression (PLSR) with the discrimination power of a classification technique [23]. The PLSR transforms the original data into new orthogonal variables referred to as latent variables (LVs). An optimal number of LVs can be determined by cross-validation. In this study, the performance was evaluated by 5-fold cross-validation, i.e., the calibration model was built with 80% of the samples to determine the optimum parameter of classification, and the rest served as test set. This procedure was repeated 10 times to obtain predictions. The PCA and the PLS-DA was conducted on MATLAB R2017b (The Mathworks, Natick, MA, USA) using in-house routines.

## 3. Results and Discussion

### 3.1. Characteristics of UV–Vis Spectra

Olive oil possesses a unique assignment of the major visible absorption bands. Its visible spectra characteristics of MTP200 are shown in Figure 1a. These spectral profiles were in accordance with a series of previous studies. Specifically, tocopherol, which has maximum absorbance peak at 325 nm [24], was observed by expanding the spectra using an approaching zoom. Concerning the olive oil visible spectra, the first peak appears at about 420 nm. This area corresponds to the absorption by olive oil of dark blue colored light, which could mainly be due to carotenoids, as well as to pheophytin *a*, pheophorbide *a* and pyropheophytin *a* [25]. The second peak was near 460 nm and also attributed to the absorption of blue light, which is characteristic of carotenoids: *α*-, *β*-, and *γ*-carotene at 447, 451, 462 nm, respectively [16]. The final peak was observed near 670 nm and it may agree with the absorption peak from chlorophyll [25]. However, no clear peaks could be found to characterize the origin of EVOO by mere visual inspection. Since the visible absorption bands indicated complex nature of UV–Vis active compounds, chemometrics modelling was examined further.

### 3.2. Comparison of Spectra between Microtiter Plate Reader and Benchtop Spectrophotometer

Since the microtiter plate reader was applied in the quality assessment of food ingredients, it is important to investigate whether the spectra obtained by the microtiter plate reader is equivalent or similar to the counterparts obtained by the benchtop UV–Vis spectrophotometer. The UV–Vis spectra of benchtop spectrophotometer are shown in Figure 1b. It can be observed that two UV–Vis spectral data sets have a similar absorbance or optical density performance tendency, but the absorbance intensities were different between two assays. Specifically, the spectra of benchtop spectrophotometer are around 1.5 times higher in absolute absorbances than their counterparts of the microtiter plate reader. The differences may be induced by a number of factors, one of them being the different instrument structure. Although both instruments follow the Beer-Lambert’s Law, some differences exist. The most typical difference is that in the UV–Vis spectrophotometer assay, the light path is horizontal and focuses perpendicularly to the vessel. In comparison, the path length is vertical [26] and varies with the sample volume (weight) in the microtiter plate reader method. The comparison of the two systems [27] is shown in Figure 2. As a result, the volume (weight) of the amount of analyte is one of the determinant factors in the absolute spectral response of the microtiter plate reader. Besides, other reasons that induce the spectral differences may include different electronic detectors; the inherent electronical noise that varies among different instruments; the solution containers, the plastic microtiter plate of which is usually made of transparent polyethylene materials instead of quartz cuvette, applied in the microtiter plate reader. The application of a microtiter plate allowed a rapid, high-throughput measurement of the microtiter plate reader. Therefore, the subsequent spectra may present differences.

Verification of the spectral similarity of two assays was carried out. Comparisons of the mean optical density/absorbance of all the EVOO samples along the entire wavelength range with different instruments and different volumes were performed. The correlation matrix is shown in Table 1. The first column of Table 1 consists of the correlation coefficients of the UV–Vis spectrophotometer assay and the microtiter plate reader assays at different volumes (weight). In this table, as the volume (weight) increases, the correlation coefficient becomes closer to 1. Therefore, to obtain a similar effect to the UV–Vis spectrophotometer assay, the optimal weight of EVOO is 300 mg for the microtiter plate reader measurement in this study, around only 1/10 of that of the benchtop instrument.

Since the two instruments possess different characteristics, further assessments of the spectral similarity are necessary. A typical absorbance peak at 415 nm was selected for all the subsequent calculations. First, the linearity of the microtiter plate reader response was performed by examining different sampling amounts. The average optical density of all samples and their weight showed the linear relationship
OD = 0.0054w + 0.0176.

OD stands for optical density and w for weight (w/mg), with a squared correlation coefficient (R^2^) of 0.9996, suggesting that the EVOO measured by the microtiter plate reader is also in accordance with the Beer-Lambert’s Law.

The relationship of actual linearity of pathlength and absorbance was also examined. The average density of the sample was 0.9113 g/cm^3^ and the diameter of each microwell was 0.86 cm. Therefore, the estimated pathlengths for 100, 150, 200 and 300 mg EVOOs were 1.9, 2.8, 3.8 and 5.7 mm, respectively, provided that the meniscus was flat. The pathlength of the cuvette was 10 mm, which was respectively 5.2, 3.6, 2.6 and 1.8 times greater than the microtiter plate reader measurements by 100, 150, 200 and 300 mg EVOOs. In comparison, the average absorbances in 415 nm by the UV–Vis spectrophotometer were respectively 3.1, 2.0, 1.5, and 1.0 times greater than the microtiter plate reader using 100, 150, 200 and 300 mg EVOOs. According to the Beer-Lambert’s Law, the pathlength is proportional to absorbance when the absorptivity coefficient and concentration of the sample are unchanged. The actual performance differences can be due to other factors like instrument differences, such as differences in the optical parts. On the other hand, a weight of 321.58 mg would theoretically result in a similar spectrum in the microtiter plate reader as in UV–Vis spectrophotometer, provided that the optical performances were ideally equal between the two instruments. Summing up, the result indicated that the two methods can achieve similar theoretical performance. Consequently, the microtiter plate reader can be a proper alternative when measuring the absorbance of EVOOs.

### 3.3. Pre-Processing of Spectral Signals

Pre-processing is frequently used to improve signal-to-noise before chemometrics modelling in spectroscopy [28]. The light scattering corrected spectra are shown in Figure 3a. The spectra of different EVOOs can be easily differentiated from each other at the optical density peak 415 nm in Figure 3a compared with untransformed spectra (seen in Figure 1a). The spectra were further processed by first derivative to eliminate the background constant, seen in Figure 3b. In a previous study, it was suggested that the proper pre-processing approach was hard to assess prior to model evaluation, and the test of different pretreatments was necessary in order to select the most suitable one [8]. Therefore, different pre-processing methods were evaluated by the complete classification process to select the optimal methods. Specifically, commonly used denoise methods in spectral data processing, including the Savitzky-Golay (SG) smoothing method, the second derivatives, and the extended multiplicative scatter correction (EMSC) [16,18,29] were evaluated. However, they showed no significant improvement in the further prediction models, compared with the original spectral data (data not shown). Therefore, the MSC and the first-level derivatives which generate the most successful PCA and PLS-DA models were applied to the spectra.

### 3.4. Classification of Producing Region of EVOOs by Microtiter Plate Reader Spectra

A PCA was performed on the MSC-Der1 processed MTP200 dataset. First, the optimal number of PCs was determined before analysis. The first four PCs were accounted for 96.28%, 2.47%, 0.94%, and 0.23% of the total variance, suggesting a model with two PCs is proper to describe the overall model variance. However, it is unprecise to solely regard a small variance captured as background noise, given that such variance may bring some information regarding the small quantities of a chemical compound [16]. Therefore, additional evaluations were required. The PCA loadings bring additional information that can help determine the proper PC number. When a PC loading is close to the baseline without apparent peaks, it might be possible to infer that this PC is not informative. The PCA loadings plot of MSC-Der1 processed MTP200 data is shown in Figure 4, in which the loadings of PC3 and PC4 were close to background noise. Therefore, the appropriate number of PCs was determined to be 2, based on both the explained variance and the loadings plot.

Figure 5a,b show scores plots of the first two PCs of MSC-Der1 processed spectra data and untransformed data, respectively. In Figure 5a, the Turkish olive oil can be clearly separated from the counterparts originated in the other two regions, but it was not possible to visually discriminate the Italian olive oil from the Spanish counterparts. However, the score plots in Figure 5b presents a remarkable improvement in the discrimination of Italian and Spanish olive oils. The clear graphical clustering in scores plots demonstrates the MSC-Der1 pretreatment of spectra data improved the PCA model predictions.

The classification model was built by PLS-DA on the MTP200 dataset. The numbers of latent variables for three class PLS-DA and the corresponding prediction accuracy were calculated. The prediction accuracy reached 97.92% in PLS-DA model built using five latent variables, indicating the feasibility to classify olive oils from different regions using the microtiter plate reader spectra. The prediction accuracy did not increase greatly as the number of latent variables increases further. The best classification accuracy was 99.58% using models built by 10 latent variables, however the high number of latent variables was not preferred because it may lead to overfitting. In summary, both PCA and PLS-DA models had adequate discrimination power in differentiating the producing regions of EVOO.

### 3.5. Quality Assessment between EVOOs and Other Vegetable Oils

An experiment was further carried out by expanding the current sample set to examine the capability of the microtiter plate reader in the quality assessment of EVOOs. The spectral data of 8 vegetable oil samples were added into the original spectral data set. Some spectra of vegetable oils contain overflown optical density readings. The overflown intensities were replaced by 4, which is the maximum indication of the microtiter plate reader. All other data-processing methods were kept unchanged from those of the original data set. From Figure 6, it was plausible to visually discriminate spectra representing different types of oil. The reason might be that the chemical characteristics of vegetable oil is relatively more heterogeneous from that of olive oil. As was shown by only using the first two PCs scores in Figure 7, the microtiter plate reader was suitable not only to discriminate the origin of EVOOs, but also to classify EVOOs from other common vegetable oils. 

## 4. Conclusions

A high-throughput and rapid UV–Vis spectroscopic method using the microtiter plate reader was examined to differentiate the producing region of EVOOs. The microtiter plate reader offers high-throughput measurement with a relatively small sample volume requirement, while maintaining adequate precision and accuracy compared to its benchtop counterparts. From the spectra comparison and statistic evaluation it can be observed that the microtiter plate reader might be a valid substitute of benchtop UV–Vis spectrophotometer. In addition, the PCA and PLS-DA are proved to be successful modelling approaches applied in EVOOs region discrimination based on the microtiter plate reader spectra. Besides, the application of MSC and first derivative treatment in spectral data improved the clustering in PCAs scores scatter plots. The result suggests microtiter plate readers may potentially be applied in many other areas of food quality assessments.This study was an initial attempt to apply the microtiter plate reader as a high-throughput spectroscopic instrument. It should be mentioned that all samples in this study were bought from the market and may lack authentication, therefore the models built in this study may not be used directly for practice. More authentic samples, such as those directly obtained from related local olive oil producers, may be needed to support food authentication reliably.

## Figures and Tables

**Figure 1 sensors-19-04169-f001:**
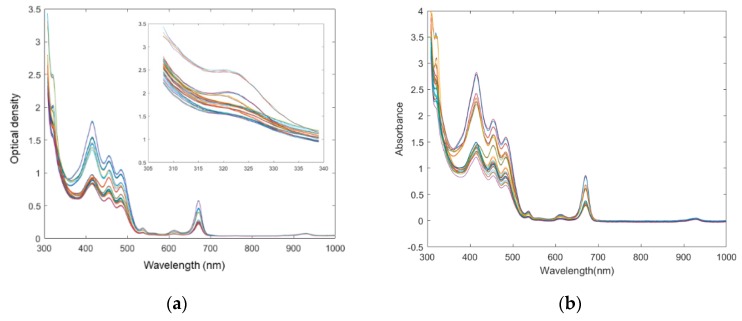
Ultraviolet-visible (UV–Vis) spectra of extra virgin olive oils from different regions obtained by different instruments. (**a**) microtiter plate reader; (**b**) benchtop UV–Vis spectrophotometer.

**Figure 2 sensors-19-04169-f002:**
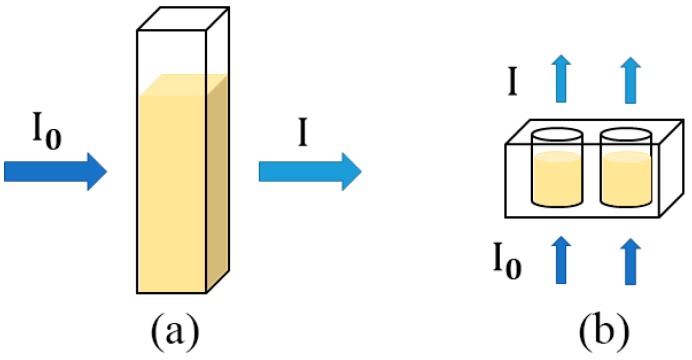
Differences of the UV–Vis spectrophotometer system and the microtiter plate reader system. (**a**) a cuvette used in the benchtop spectrophotometer and (**b**) a microtiter plate used in the microtiter plate reader system. I_0_ and I indicate the optical intensities before and after detection, respectively.

**Figure 3 sensors-19-04169-f003:**
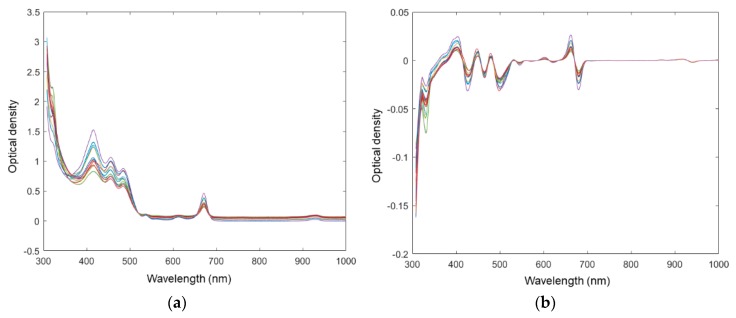
(**a**) Multiplicated scatter corrected and (**b**) multiplicative scatter correction followed by the first-level derivative (MSC-Der1) pretreated microtiter plate reader spectra.

**Figure 4 sensors-19-04169-f004:**
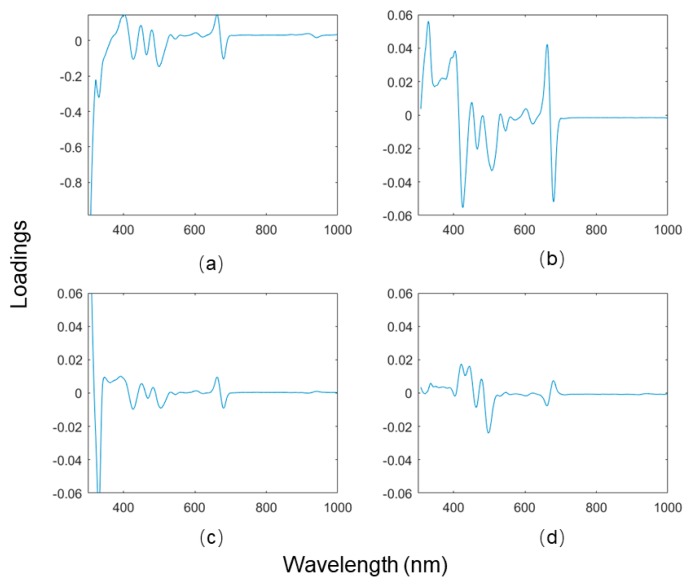
Principal component loadings plot of the first four principal components (PC1-PC4), (**a**) PC1; (**b**) PC2; (**c**) PC3; (**d**) PC4.

**Figure 5 sensors-19-04169-f005:**
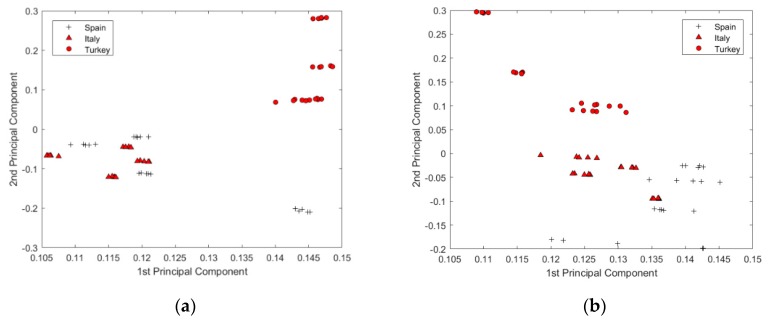
Scores plot of (**a**) untransformed raw data, (**b**) multiplicative scatter correction followed by the first-level derivative (MSC-Der1) pretreated data.

**Figure 6 sensors-19-04169-f006:**
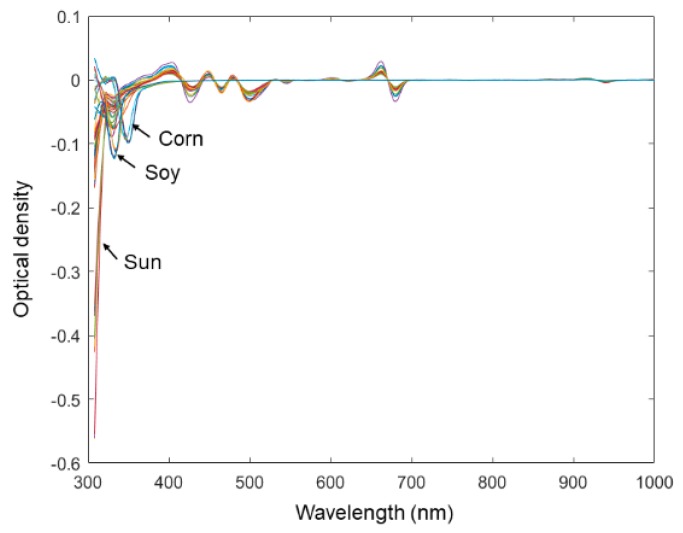
Spectra of multiplicative scatter correction followed by the first-level derivative (MSC-Der1) pretreated data of extra virgin olive oils and other vegetable oils.

**Figure 7 sensors-19-04169-f007:**
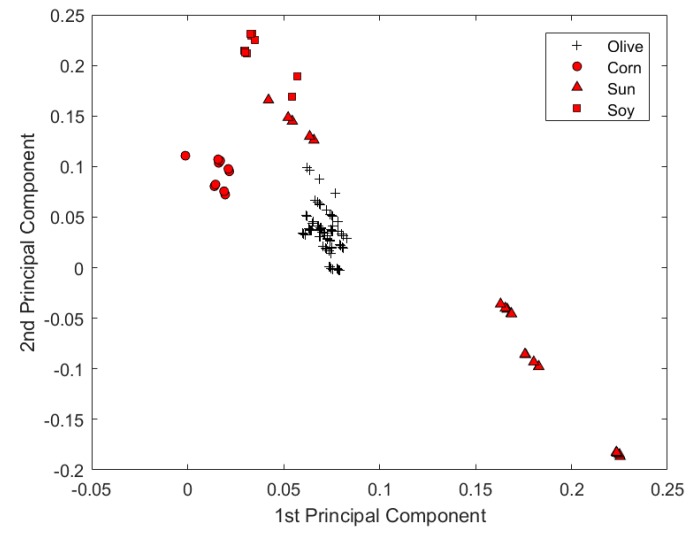
The principal component scores plot of quality assessment between extra virgin olive oils and other vegetable oils by the microtiter plate reader.

**Table 1 sensors-19-04169-t001:** Correlation matrix for optical density/absorbance of all extra virgin olive oils with different instruments and different volumes (weight) with a significance level of 0.05.

	Benchtop	MTP100	MTP150	MTP200	MTP300
Benchtop	1				
MTP100	0.9957	1			
MTP150	0.9976	0.9997	1		
MTP200	0.9988	0.9989	0.9997	1	
MTP300	0.9995	0.9973	0.9987	0.9995	1

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
