# Peer review of "High-Throughput Chemometric Quality Assessment of Extra Virgin Olive Oils Using a Microtiter Plate Reader"

_sensors, 2019, doi:10.3390/s19194169_

Round 1
Reviewer 1 Report
Authors developed a method by using a microtiter plate reader as a high-throughput UV-Vis spectrophotometer for the quality assessment of food ingredients such as extra virgin olive oil. The work is interesting, it provides a potential approach for quality assessment of food ingredients.
I have several comments listed below.
How to ensure the authenticity of selected samples. Comparison with other methods needs to be provided, preferably provided in a table. The sample size is too small to establish such a method.Author Response
Point 1: How to ensure the authenticity of selected samples. Comparison with other methods needs to be provided, preferably provided in a table. The sample size is too small to establish such a method.
Response 1: We thank you for the helpful comment to improve this manuscript. The sample authenticity is discussed in the end of the conclusion. Moreover, we have revised the related discussion in further detail. We mentioned that the models may not be used directly for practice. To ensure sample authenticity, the samples may be collected from local olive oil producers. However, more resources and efforts may be needed to complete this approach. Our main aim of this study is to demonstrate the potential of a high-throughput UV-Vis spectroscopic method, therefore the main conclusion still holds.
The comparisons of microtiter plate reader in different volume, as well as with a commercial Evolution 300 UV-Vis spectrophotometer was given in Figures 1 and 2, and Table 1 in the Results and Discussion section.
We agree to the reviewer that a large-scale study would be more meaningful in terms of increasing the statistical power. However, due to the limited resources, the number of samples collected is often limited. On the other hand, our main aim of this study is to demonstrate the potential of a high-throughput UV-Vis spectroscopic method. By observing the main clustering trend, the samples were separated by both PCA and PLS-DA with clear discrimination power. Consequently, the model is adequate for demonstrating the possibility of a microtiter plate reader applied in some typical food authentication and quality assurance.
Reviewer 2 Report
The paper presents a new approach for analysis of food product. The obtained results have a chance to find applications in food control industry section. The paper is prepared correctly. The content is easy to follow. I recommend this paper for publication as it is.
Author Response
Thank you for your comment.
Reviewer 3 Report
The paper describes the application of commercially available microtiter plate reader combined to chemometrics for the discrimination of extra virgin olive oil geographical origin. Also, the proposed method was able to detect olive oil fraud using other oil types and was benchmarked towards a classic UV-Vis spectrophotometer.
Although the method may find even commercial application (it is fast, with minimal non-destructive sample preparation and cost-efficient instrumentation), the number of tested samples is rather insufficient (12 samples= 4 samples x 3 countries in the case of geographical origin discrimination & 8 samples for fraud detection). Except the low number of samples, the authors state in conclusion that “all samples in this study were bought from the market and may lack of authentication”. In conclusion, the proposed method is useful but in order to be published more authentic samples have to be tested.
Specific comments.
L10: this abbreviation is not so common. maybe is better not to use it
L19: confusing sentence. Please rephrase
L21: you may add some more keywords
L35: You may cite the following reviews to provide a better background of the field
https://doi.org/10.1016/j.trac.2016.02.026 https://doi.org/10.1016/j.tifs.2019.02.025L44: confusing sentence. Please rephrase. Also, change fingerprint to fingerprinting
L54: “to authenticate” sounds strange. May use "to verify"
L57: how do you mean alleged? Did they assume the origin in this study?
L83: Three countries
L84: and Turkey
L86: How are you sure about the origin of the samples? Did you buy them directly from producers? Please clarify how you acquire the samples
L98: why you didn't filter the samples to avoid any particles that can be a problem for the spectroscopic detection?
L158: the first peak appears about 420 nm
L163: “origin” instead “origination” would be better
L165: It would be helpful to provide a legend to indicate which spectrum corresponds to which sample
L187: Please mention also the inherent electronical noise that varies among different instruments
L237: rephrase to “it was suggested”
L245: Please rephrase the sentence, it's confusing
L306: authentic instead authenticated
Author Response
L10: this abbreviation is not so common. maybe is better not to use it
Response: Revised as suggested. the abbreviation “MTPR” is removed. Thank you for your helpful comment.
L19: confusing sentence. Please rephrase
Response: Revised as suggested. the sentence is rephrased to: “It is demonstrated that the microtiter plate reader can be a high-throughput tool in the quality assessment of food ingredients.”
L21: you may add some more keywords
Response: Thank you for your comment. According to the journal’s guideline, only a maximum of five keywords is allowed. Therefore, no changes were made.
L35: You may cite the following reviews to provide a better background of the field
https://doi.org/10.1016/j.trac.2016.02.026
https://doi.org/10.1016/j.tifs.2019.02.025
Response: Revised as suggested. The corresponding references were added in this revision.
L44: confusing sentence. Please rephrase. Also, change fingerprint to fingerprinting
Response: Revised as suggested. This sentence is revised as: “In comparison, the spectroscopic methods such as ultraviolet-visible (UV-Vis), near-infrared (NIR) and mid-infrared (MIR) spectroscopies advantages in simple, rapid, nondestructive, and low cost. As a result, they may be suitable for fingerprinting.”
L54: “to authenticate” sounds strange. May use "to verify"
Response: Revised as suggested (L55).
L57: how do you mean alleged? Did they assume the origin in this study?
Response: Yes, they assumed the origin. Specifically, the olive oil samples in the study by Alves et al. were purchased in the marketplace without authentication. We keep the word “alleged” as they used originally in their manuscript. Therefore, no changes were made in this revision.
L83: Three countries
Response: Revised as suggested (L83). Thank you for your comment.
L84: and Turkey
Response: Revised as suggested (L84). Thank you for your comment.
L86: How are you sure about the origin of the samples? Did you buy them directly from producers? Please clarify how you acquire the samples
Response: Thank you for your valuable comment. As stated in Section 2.1 (L83) Twelve commercially available EVOO samples were purchased from local Chinese grocery stores. Such approach has been used in other studies such as the one by Alves et al.(Ref. 16). The sample authenticity is also discussed in the end of the conclusion (L309). We agree to the reviewer that sample origin may cause problem when applied directly, therefore, we have the related discussion in further detail at the end of the Conclusion. We mentioned that the models may not be used directly for practice. To ensure sample authenticity, the samples may be collected from local olive oil producers. However, more resources and efforts may be needed to complete this approach. Our main aim of this study is to demonstrate the potential of a high-throughput UV-Vis spectroscopic method, therefore the main conclusion still holds.
L98: why you didn't filter the samples to avoid any particles that can be a problem for the spectroscopic detection?
Response: Mathematical pretreatments such as standard normal variate (SNV), multiplicative scatter correction (MSC), derivative methods and orthogonal signal correction (OSC) are proved by other researchers and more common in decreasing the interference by particles to the spectra data (L115, Ref. 8).
L158: the first peak appears about 420 nm
Response: Revised as suggested (L159).
L163: “origin” instead “origination” would be better
Response: Revised as suggested (L164).
L165: It would be helpful to provide a legend to indicate which spectrum corresponds to which sample
Response: Thank you for your comment. We have tried your suggestion to put a legend, but it is overcrowded and viewed poorly. We decided to keep the original figure in order to keep the presentation of data here concise and clear. However, we added data availability statements, so that anyone who is interested in the detail of the results can further view it in their own manner.
L187: Please mention also the inherent electronical noise that varies among different instruments
Response: Revised as suggested. The statements of electronical noise were added correspondingly.
L237: rephrase to “it was suggested”
Response: Revised as suggested (L241).
L245: Please rephrase the sentence, it's confusing
Response: Revised. “Chemometrics analyses” were changed to “Classification of Producing Region of EVOOs by Microtiter Plate Reader Spectra” Thank you for your valuable comment to improve this manuscript.
L306: authentic instead authenticated
Response: Revised as suggested (L312).
Round 2
Reviewer 1 Report
I have no other comments. If possible, I still recommend increasing the sample size.
Reviewer 3 Report
The authors applied the proposed comments resulting in an improved manuscript. Regarding the authenticity of commercial samples(which was my major argument), the authors clearly stated in Conslusions that a greater authentic sample-set is needed to further demonstrate the study applicability. Thus, i propose to accept the study in its current form after rechecking minor text editing problems (for example L45-L47, L322-L324)